# Low Temperature Microcalorimeters for Decay Energy Spectroscopy

**Katrina E. Koehler** 

Los Alamos National Laboratory, Los Alamos, NM 87545, USA; kkoehler@lanl.gov

**Abstract:** Low Temperature Detectors have been used to measure embedded radioisotopes in a measurement mode known as Decay Energy Spectroscopy (DES) since 1992. DES microcalorimeter measurements have been used for applications ranging from neutrino mass measurements to metrology to measurements for safeguards and medical nuclides. While the low temperature detectors have extremely high intrinsic energy resolution (several times better than semiconductor detectors), the energy resolution achieved in practice is strongly dependent on factors such as sample preparation method. This review seeks to present the literature consensus on what has been learned by looking at the energy resolution as a function of various choices of detector, absorber, and sample preparation methods.

**Keywords:** decay energy spectroscopy; Q spectroscopy; microcalorimeter; transition edge sensor; metallic magnetic calorimeter; thermistor; embedded; absorber



## 1. Introduction

Low temperature detectors have been used for various applications [1] ranging from low energy measurements like X-rays to gamma-rays to high energy measurements of $\alpha$ particles and even fission fragments. Excellent reviews of the X-ray [2–5] and gamma-ray applications [3] exist and there are few low temperature measurements of $\alpha$ particles [6–11] and fission fragments [12,13], so those papers can be easily read for an entire review of the field. Low temperature detectors have also been used in the internal measurement mode with macrocalorimeters and bolometers, but this review focuses on microcalorimeters, which measure thermalized energy on an event by event basis. Although the use of microcalorimeters for the internal or embedded technique has three decades worth of publications, a review of the technique is lacking, particularly as it pertains to the energy resolution achievable with this technique. Low temperature detectors' primary advantage is superior energy resolution as compared to semiconductor detectors. For gamma-rays, an order of magnitude resolution better than High Purity Germanium detectors is standard at 100 keV (resolving power of about 1500) and for X-rays the story is similar. For $\alpha$ detectors, achieving less than 1 keV FWHM on a 5 MeV peak is typical with careful consideration to source preparation and position sensitivity of the detector, which yields a resolving power of about 5000. For fission fragments, achieved resolution is now 91 keV for measurements of 20.7 MeV ions [12]. However, for internal measurements with Decay Energy Spectroscopy, the achieved resolution was falling short of the predicted achievable resolution.

Decay Energy Spectroscopy (DES) is a particular measurement mode using low temperature detectors where the radioactive source is embedded within an absorber. This absorber thermalizes the decay energy and is coupled to a low temperature detector, such as a Transition Edge Sensor (TES), Metallic Magnetic Calorimeter (MMC), or other low temperature detector. When the source decays, the decay radiation (i.e., electrons, X-rays, gamma-rays, $\alpha$ particles, recoil daughter nuclei) is thermalized within the absorber. Due to the long thermal time constants of DES detectors, the entire energy of the decay is measured

rather than the energies of individual particles involved in the decay. This is excepting the radiation that escapes the absorber, such as neutrinos and high energy gamma-rays. The detector becomes a highly efficient method of measuring radiation and can be considered a $4\pi$ detector if careful attention is paid to whether material is on the surfaces of the absorber or entirely contained within the volume of the absorber. As a result, these detectors can be used for very low activity measurements, often targeting 1–10 Bq. Due to the long time constants of these detectors, higher activities present a challenge of pileup.

Aside from the very high energy resolution that comes from using low temperature detectors, the DES measurement technique offers another significant advantage: it is independent of decay path. For example, measuring the $\alpha$ energy spectrum of an isotope that $\alpha$-decays results in peaks for each $\alpha$ energy. Measuring the same isotope with DES results in a single peak corresponding to the $Q$ value or reaction energy. This is because the detector is able to thermalize energy regardless of its form, such as kinetic energy in the recoil daughter nucleus or gamma-rays resulting from an $\alpha$ decay to an excited state of the daughter nucleus. Provided the energy does not get stored in a long-lived state, the energy will be read out by the DES detector and can simplify many complicated spectra resulting from a variety of decay paths.

This review aims to highlight the diversity of applications of the DES technology, giving a brief history of Decay Energy Spectroscopy over the past few decades. In particular, the review is focused on the discussion of achieved versus predicted resolution for these detectors. This review will discuss the probable cause and solutions as given by the body of literature on DES.

## 2. Materials and Methods

A internet search of papers using the Google Scholar search engine was performed with the following search terms: "Q Spectroscopy" AND "low temperature", "Decay Energy Spectroscopy", "Decay Energy Spectroscopy" AND "low temperature", "Transition Edge Sensors" AND embed, and "Metallic Magnetic Calorimeters" AND embed. Search terms like "microcalorimeter" and "thermistor" were considered, but these searches did not provide any new results not already captured by the other searches. The results were winnowed to include only those papers that were potentially of relevance to this review. For example, not all microcalorimeters are low temperature devices, defined as operating at temperatures lower than liquid He (<4.2 K). This winnowing led to a total of 92 published papers. Dissertations and programmatic reports were not included. As these papers were analyzed, it was found that some papers only referenced other papers on decay energy spectroscopy, but had no DES results contained in them. Only papers including DES spectra or resolution values were included in the analysis, leaving 40 papers to analyze. While this method may not result in 100% of the papers on the topic, it is expected to be representative of the DES literature.

Each paper was tagged with certain metadata as well as information from the article itself. A few representative papers and their tags can be seen in Table 1. For each reference, the reported resolution was included. On the rare occasion that a spectrum was shared without a resolution reported, the FWHM was estimated from the spectrum. Since not all detector systems have the same energy resolution limits, information on the achievable resolution was also included. These values came from the same paper. For example, the DES measurement of electron capture might have a particular resolving power, but an X-ray measurement with the same detector might show a better resolving power. These values when available were included, such as those shown in Table 2. Some papers were vague about where predicted values were from and merely mentioned that "tests" were performed or the values came from knowledge of the detector, such as [14].

**Table 1.** Meta data for four papers. These data were derived from information within the paper where possible.

| Ref. | First Author | Year | Application | Detector | Absorber Material | Nuclide | Deposition | Attachment |
|------|-------------|------|-------------|----------|-------------------|---------|------------|------------|
| [15] | Cosulich | 1992 | Neutrino mass | NTD-Ge | Re | $^{187}$Re | natural | |
| [16] | Gatti | 1997 | Neutrino mass | NTD-Ge | Sn | $^{163}$Ho | dried drop | epoxy |
| [14] | Deptuck | 2000 | Neutrino mass | TES | Cu | $^{3}$H | ion implantation | |
| [17] | Voytas | 2002 | Atomic Theory | Si:P:B | Si | $^{7}$Be | ion implantation | epoxy |

**Table 2.** Energy resolution values for four papers. The resolving power is determined by dividing the energy by the FWHM. All energies are in units of keV.

| Ref. | Radiation Source | FWHM | E | Power | Comparison Source | FWHM | E | Power |
|------|-----------------|------|---|-------|-------------------|------|---|-------|
| [15] | Beta | 0.05 | 1.5 | 30 | X-ray | 0.565 | 30 | 53 |
| [16] | Beta | 0.08 | 6 | 75 | | | | |
| [14] | EC | 0.0085 | 0.12 | 14 | tests | 0.03 | 6 | 200 |
| [17] | Beta | 0.09 | 62 | 689 | X-ray | 0.006 | 5.9 | 983 |

## 3. History

A review of these articles allowed for a reconstruction of the history of DES. The idea of using microcalorimeters in an internal measurement mode, where the source is embedded within the absorber rather than being external to it, has been explored since 1992. The history of spectroscopic measurements with embedded sources has been plagued with less-than-expected energy resolution, and this is shown in Figure 1. The history is recounted below to make the important point of the difficulty in embedding sources in such a way as not to degrade energy resolution in the process. Even after the best energy results are achieved, subsequent measurements fail to achieve this standard because applications, detectors, absorbers, deposition methods, and sample preparation methods differ by institution. Methods that work for one particular application may be physically impossible, impractical, or too expensive to repeat for a different application. For this reason, the space mapped out by these variables is explored in this review.

### 3.1. Low Energy DES Measurements

DES measurements began with low energy measurements (<2 MeV) of radionuclides decaying primarily by electron capture and beta decay. Beta decay measurements with embedded $^{187}$Re for the purpose of neutrino mass measurements were the first explorations of this type of measurement. The advantage here being that Re is a superconducting material at around 1.7 K [18], with the isotope of interest, $^{187}$Re, naturally occurring in metallic Re at 63%. After initial work on understanding the metallic rhenium crystal properties, the MANU Re beta experiments in Genoa reported the first embedded spectrum in 1992 as measured with a Neutron Transmutation Doped (NTD) thermistor [15]. MANU discovered the Beta Environmental Fine Structure (BEFS), a result of self-interference of the outgoing beta-waves with the reflected beta-waves from surrounding atomic shells [18]. This manifested as an undulating modulation on the spectrum.

In 1992, the MIBETA Re beta experiment in Milan was begun to work on the $^{187}$Re experiments, employing eight silicon-implanted thermistors coupled to four crystals of silver perrhenate ($AgReO_4$) [19–21]. MIBETA took data between 2002 and 2003. Both MANU and MIBETA were able to provide bounds on the direct anti-neutrino mass of about 26 eV at 95% CL and 15 eV at 90% CL, respectively.

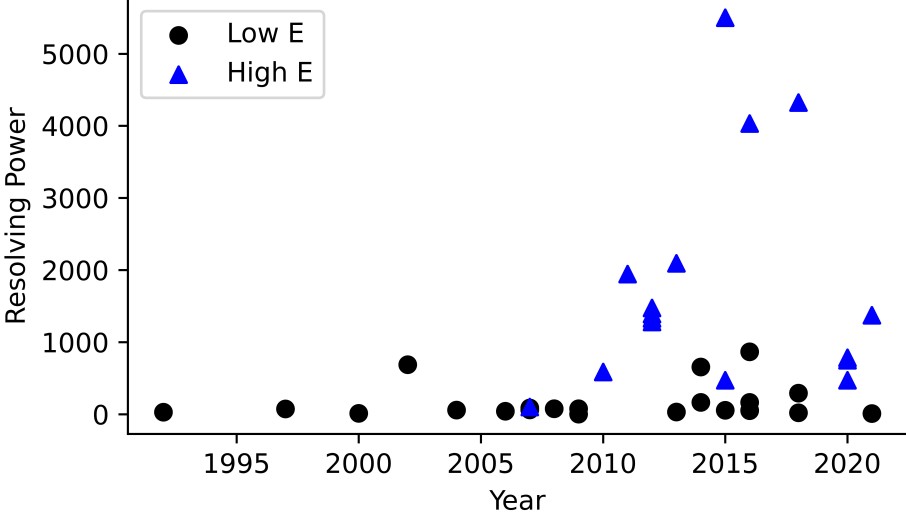

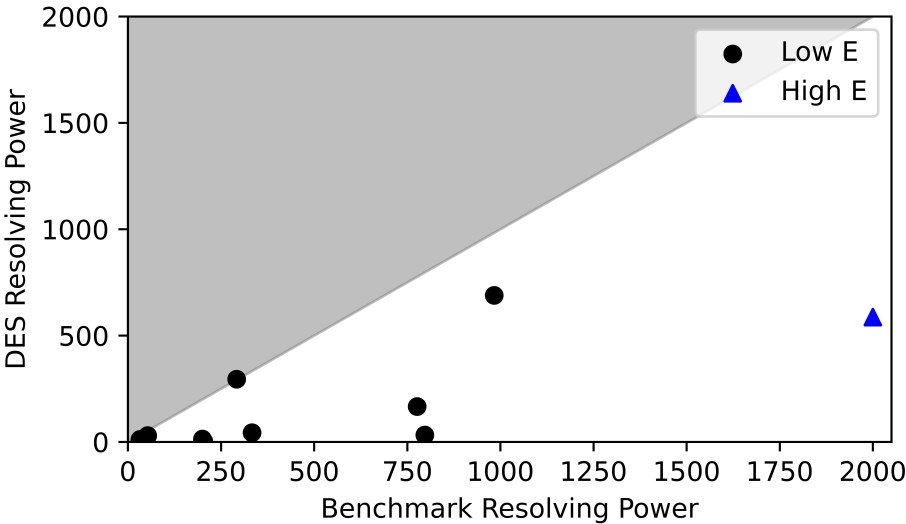

**Figure 1.** (*Top*) Detector resolving power versus publication date. The first high energy (>2 MeV) DES results occur in 2007 and achieve results exceeding a resolving power of 5000 by 2015. For the low energy (<2 MeV) measurements, the best measurements achieve resolving powers less than 1000. (*Bottom*) The resolving power of DES measurements is compared to benchmark measurements, which correspond to predictions based on the noise or baseline signal, measurements of other radiation such as X-rays or gamma rays, or from prior detector knowledge. In this case, some measurements with very low resolving power are predicted and achieved, but predictions of higher resolving power as is standard with low temperature microcalorimeters were frequently not achieved in these DES measurements.

The Microcalorimeter Arrays for a Neutrino Mass Experiment (MARE) was formed toward the end of these two experiments in 2006, with a mandate to explore $^{187}$Re as a potential source and various low temperature microcalorimeters for a direct neutrino mass measurement [22–24]. There were early indications that the Re absorber was not behaving as expected for a superconducting absorber [25], with measured heat capacity of the absorbers several orders of magnitude higher than superconducting Re and long thermal tails on the pulses. After a decade of attempts to make Re work as an absorber, the project eventually turned to $^{163}$Ho as the best candidate, having never succeeded in improving the energy resolution sufficiently [26]. The results of MARE were similar to those of its predecessors: "Rhenium absorbers behave inconsistently, showing a large

deficit in the energy thermalization accompanied by long time constants" [27]. A good review of the history of rhenium absorbers is given in [27].

In the meantime, in 1997, Gatti et al. measured [163]Ho within a tin absorber coupled to an NTD Ge thermistor. The holmium source was dried onto a tin foil from a water solution and covered with epoxy. The foil was then folded on itself and pressed to form a thermal link with the epoxy. The results of this measurement were lower than expected energy values of the measured peaks, asymmetric peaks, and degraded resolution. All of these effects were attributed to inefficient energy thermalization due to the inhomogeneous absorber (epoxy and tin) [16].

In the early 2000s, a brief foray into using TESs to measure the beta spectrum of tritium was conducted [14,28], but the detectors ended up being too slow to achieve meaningful results for neutrino mass measurements, as had been the purpose.

From 2003–2008, Cuoricino used bolometers to measure double beta decay on [130]Te within $TeO_2$ crystals and actually achieved better energy resolution than predicted [29]. This experiment developed into the CUORE neutrinoless double beta decay experiment with a better sensitivity for neutrinoless double beta decay [30]. The bolometer research is not included in the papers analyzed in this review article due to mostly being used in macrocalorimeter research.

In 2008 and 2010, Loidl et al. published a couple papers using the embedded source method for metrological purposes. The radionuclides of interest were dissolved in a solution and dried onto a gold foil, which was folded over, pressed, and diffusion-welded to encapsulate the material, then coupled to a Au:Er MMC [31,32]. In the first measurement [31], the spectral resolution allowed for differentiation between the K captures and L captures of [55]Fe, but was not sufficient to differentiate between the electronic subshells (i.e., capture from the L1 shell versus L2). Energy loss within the source ($FeCl_3$ crystals) contributed to spectral distortions in this measurement and there was a substantial difference in the detector response to X-rays versus Auger electrons. While the resolution was far from optimal, for the purposes of measuring the total capture probabilities, the achieved resolution was sufficient. In the second paper, the same measurement technique with a diffusion-welded foil containing a dried sample was used to measure the beta spectrum of [241]Pu [32]. There was some question with this measurement about whether other material co-deposited with the source (specifically a substance from the chemical separation of Pu and Am) contributed to spectral changes arising from different thermalization between electrons and photons. In 2017, a second measurement of [55]Fe featured an electroplated sample on a gold foil that was subsequently diffusion-welded at 400 °C [33]. This measurement was successful in both cleaning up the spectrum and lowering the energy threshold to 50 eV to include the M captures.

About this time, direct neutrino mass experiments, such as Electron Capture [163]Ho experiment (ECHo), Holmium neutrino experiment (HOLMES), and Neutrino Mass via [163]Ho Electron Capture Spectroscopy (NuMECS) explored methods of embedding [163]Ho in gold absorbers. For excellent reviews of neutrino mass measurements with [163]Ho, see [27,34].

For NuMECS, the [163]Ho in a HCl solution, was dried onto a gold foil or within a nanoporous gold layer [35]. The former underwent the same mechanical kneading process as had prior samples. Despite this, poor resolution and different pulse shape from the electroplated [55]Fe sample within the same absorber prompted the team to pursue a Au nanofoam absorber. The nanofoam's small pore sizes (50–100 nm) limit the size of crystalline structures formed during the deposition of material. A second step of annealing in forming gas to remove impurities was used to improve the source-absorber matrix. These two steps greatly improved the observed spectral resolution to 35 eV FWHM at 2 keV (resolving power of 57), but the resolution still appeared to be limited by energy thermalization differences within the absorber.

HOLMES is also pursuing the implantation route using a custom ion implanter with mass separator. This has yet to be demonstrated, but the idea is that with the capability of

simultaneous gold evaporation, both initial and final layers of gold to encapsulate the [163]Ho can be laid down within the same instrument and gold can be evaporated simultaneously as the ion implantation to dilute the [163]Ho concentration [36].

ECHo is pursuing laser ion implantation within a Au absorber layer and then final encapsulation with a Au layer [37,38]. While stating that energy resolution was not degraded from the ion implantation technique, ECHo still reports energy resolutions worse than those of external [55]Fe X-ray measurements [39,40]. This is potentially due to lattice damage effects, increase in heat capacity due to the implantation process, or unequal operating parameters described in [39]. Subsequent measurements show that the spectral resolution is more than a factor of two greater than that expected from the detector's thermodynamic properties and the readout noise [41].

In coincidence with the neutrino mass measurement themes, the importance of understanding the weak decays and the atomic and molecular effects on these transitions was realized. As a result measurements were made on [14]C, [151]Sm, [99]Tc, and [36]Cl as part of the MetroBeta project exploring the beta shape [42,43]. At Los Alamos National Laboratory, measurements of [193]Pt were done to understand the atomic effects on electron capture [44]. The European collaboration, MetroMMC was created to explore electron capture [45]. At Lawrence Livermore National Laboratory, the [7]Be measurements first performed in 2002 with Si:P:B thermistors [17] were revisited in 2020 with Superconducting Tunnel Junctions [46,47], adding to the body of literature on atomic effects on nuclear weak decay.

### 3.2. High Energy DES Measurements

In 2007, the high energy (>2 MeV) measurements began. The Korea Research Institute of Standards and Science (KRISS) team used MMCs to measure the $\alpha$-decaying isotopes of [241]Am and [226]Ra using the method of drying a solution onto a gold foil followed by diffusion-welding [48–51].

Los Alamos National Laboratory (LANL) efforts on high energy DES spectroscopy began with $\alpha$-decaying isotopes [241]Am, [238]Pu, [209]Po, and [210]Po using TESs [9]. Following similar methods to those used previously, the actinide sources were dried onto a foil that was then folded and diffusion-welded. The Po source was autodeposited; however, because Po is volatile, the diffusion-welding step was omitted. Peak-splitting in the spectra from the Am and Pu measurements indicated differences in thermalization, which were eventually attributed to the material co-deposited with these solutions because these features were not seen in the Po measurements or subsequent measurements with electroplated sources [9,52,53]. The KRISS team also encountered this problem with later measurements and modeling of [226]Ra [54]. Researchers from the French Atomic Energy Commission (CEA) also found that, while their [210]Po measurements did not exhibit peak-splitting, there was an unusually long tail associated with the peak [55]. A 'tail' is a non-Gaussian feature in the spectrum, usually on the low-energy side of a peak, that is often described by an exponential. This can be due to incomplete thermalization within the absorbers.

At LANL, peak-splitting was largely eliminated with mechanical kneading or alloying of the absorber to break up the crystalline structure of the co-deposits [53]. The diffusion-welding step was also found to be unnecessary. Peak broadening and tailing in these spectra was attributed to lattice damage, or kinetic energy converted to potential energy by dislocating atoms in the crystal lattice of the absorber, similar to that seen in $\alpha$ spectroscopy with similar detectors [56]. With the method of mechanical kneading, precise measurements of the [240]Pu/[239]Pu ratio and the [241]Am/[239]Pu mass ratio could be made [53,57,58]. Additionally, simultaneous $\alpha$-decay and $\beta$-decay were measured with a [241]Am and [241]Pu source [52]. More recently these methods have been used for measurements of medical nuclides such as [225]Ac and contaminant [227]Ac [59].

## 4. Results

Decay Energy Spectroscopy has been used with a number of different cryogenic microcalorimeters (see Figure 2), with half the research having been done with MMCs [25,39–42,48–51,55,60–64] and a quarter having been done with TESs [9,14,35,35,44,52,53,57–59]. NTD-Ge thermistors were prevalent in the early research measuring $^{187}$Re for neutrino mass measurements. The Superconducting Tunnel Junction (STJ) and Si:P:B thermistor work were measurements of $^7$Be [17,46,47]. When the resolving power is plotted as a function of detector type, it becomes clear that the energy resolution is not a function of detector type, but rather with development, a detector can achieve very good resolving power. Since most development in DES measurements has been done with MMCs and TESs, both of these technologies have achieved the high resolution results expected for both high energy and low energy applications.

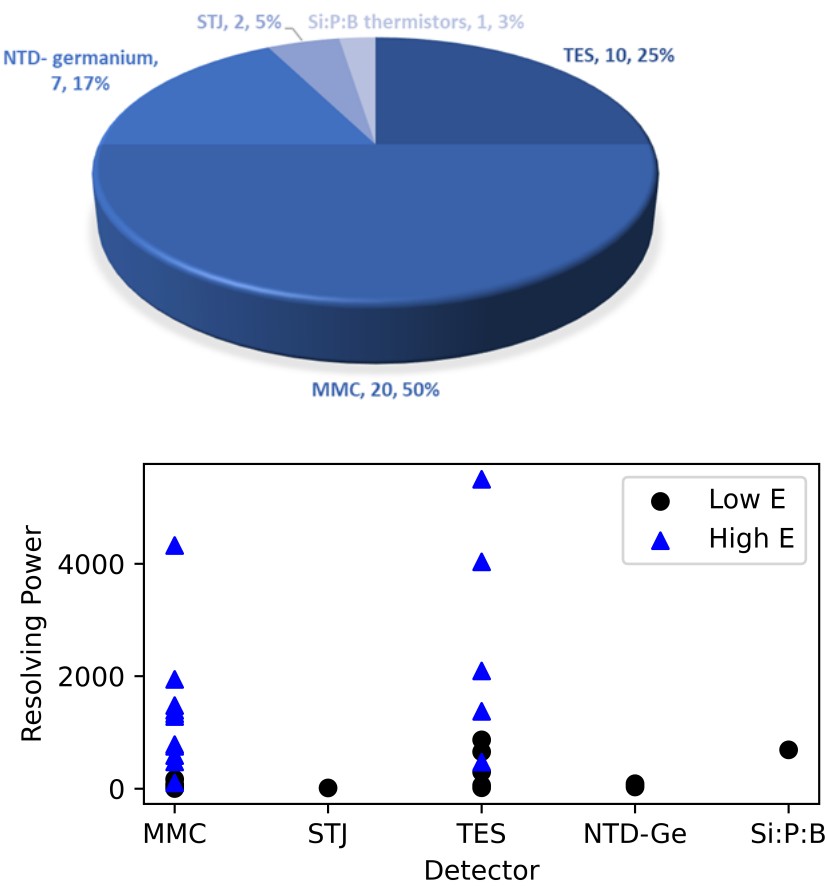

**Figure 2.** (*Top*) Prevalence of detector use for DES measurements. (*Bottom*) The resolving power is given as a function of detector and it is clear that high energy resolution measurements are not the sole domain of a single detector type.

DES detectors were first used for neutrino mass measurements and have continued to be used for neutrino physics with half of the papers on this topic [14–16,19,21,25,35, 39–41,46,47,52,61,63,65,66]. A third of the papers deal with isotopic determination for safeguards [9,50,51,53,55,57,58,62,64]. The papers presenting details on atomic theory [17,42,44,60], for environmental purposes [48,49], and medical purposes [59] comprise the remainder of the applications as can be seen in Figure 3. Some of these have overlap, most notably the low energy measurements for atomic theory and for neutrino physics.

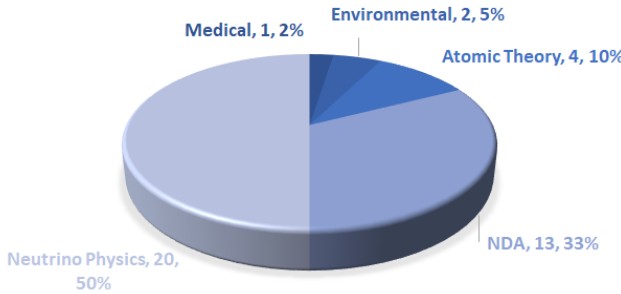

**Figure 3.** Applications of DES detectors.

The nuclides measured with DES have varied widely from low energy (<2 MeV) to high energy (>2 MeV) measurements. The former represent electron capture and beta-decaying radionuclides, while the latter are $\alpha$-decaying nuclides. These are shown in Figure 4. As a function of nuclide, the resolving power does not appear to show any discernible trend other than for each radionuclide there appears to be at least one measurement with a low resolving power, indicating that it takes multiple attempts to get a good measurement.

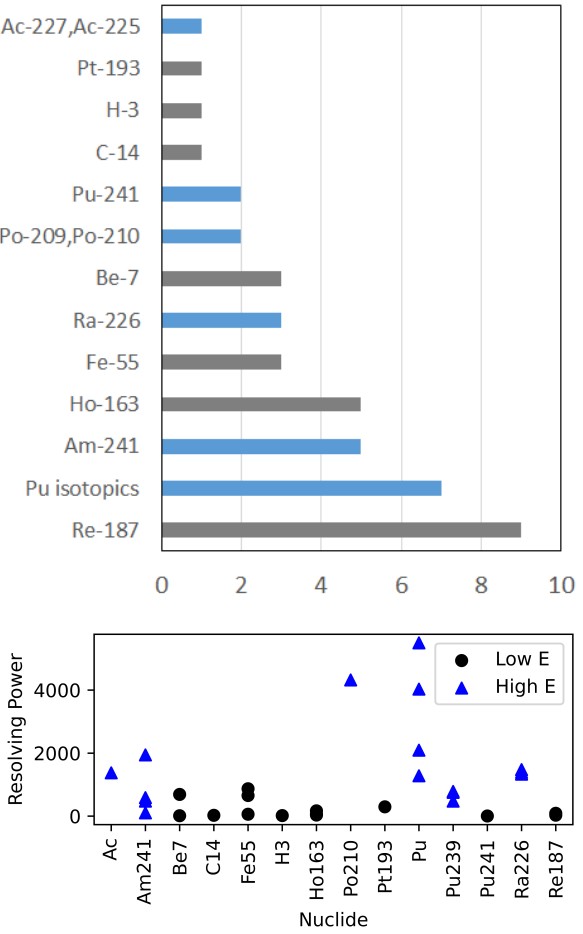

**Figure 4.** (*Top*) Nuclides measured with DES detectors. The bars in grey represent the low energy nuclides that decay primarily by electron capture or beta decay. The bars in blue represent the high energy nuclides that decay primarily by $\alpha$ decay. (*Bottom*) As a function of nuclide, the resolving power does not appear to show any discernible trend.

The majority of measurements, particularly those after the Re experiments of early 2000 s, have used Au as the absorber material as seen Figure 5. Au provides the advantage that it can be electrodeposited, which is important for encapsulation, and formed into foils, which is important for the dried drop methods of encapsulation. The Re experiments used both Re crystal and silver perrhenate (AgReO$_4$) forms, and since $^{187}$Re is found in natural Re, these did not have to be enriched in the isotope for measurement. Despite 8 measurements of the silver perrhenate form, the energy resolution never improved to the current limit of low energy DES measurements (resolving power of 1000). The Pt absorber measurement was performed with enriched and irradiated Pt, and it was determined that the heat capacity was too high for the measurements to produce high energy resolution [44]. The measurements with Si was of an ion-implanted $^{7}$Be source and the resolving power for this measurement was quite good (689) [17]. The Sn measurement was a dried drop of $^{163}$Ho in solution. This had poor energy resolution, but it is not clear if this is due to the Sn absorber or the dried droplet method [16]. The Ta absorbers were measurements of ion-implanted $^{7}$Be and achieved resolving powers on the order of 10, which is about a third lower than predicted from laser measurements [46,47]. Other approaches under investigation include using bilayer absorbers to reduce the effect of bremsstrahlung [67].

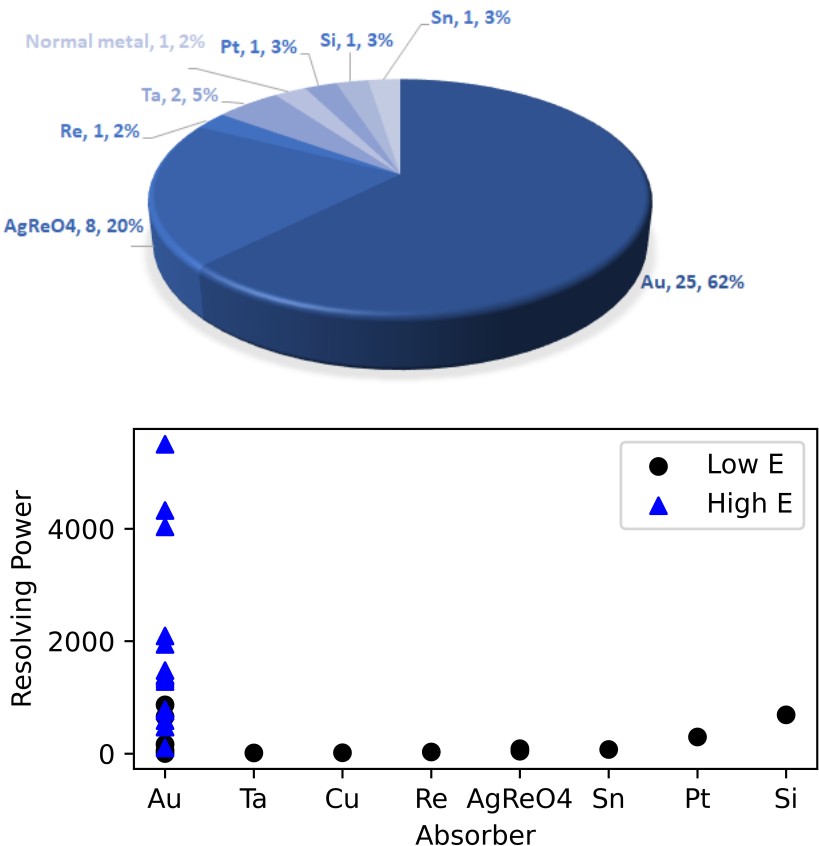

**Figure 5.** Absorber materials used in Decay Energy Spectroscopy. The resolving power can be seen as a function of absorber material for both high energy DES and low energy DES measurements.

The DES absorber is usually fabricated separately from the detector. Because of this, it must be thermally coupled to the detector for measurements. There are a variety of ways this has been done as shown in Figure 6. The most common ways are using an In bump bond or epoxy. The epoxies used included both Stycast 1266 and Stycast2850FT. The best achieved resolving power for high energy DES measurements was done with an In bump bond. For low energy DES measurements, the best resolution achieved was accomplished with both the In bump bond and epoxy.

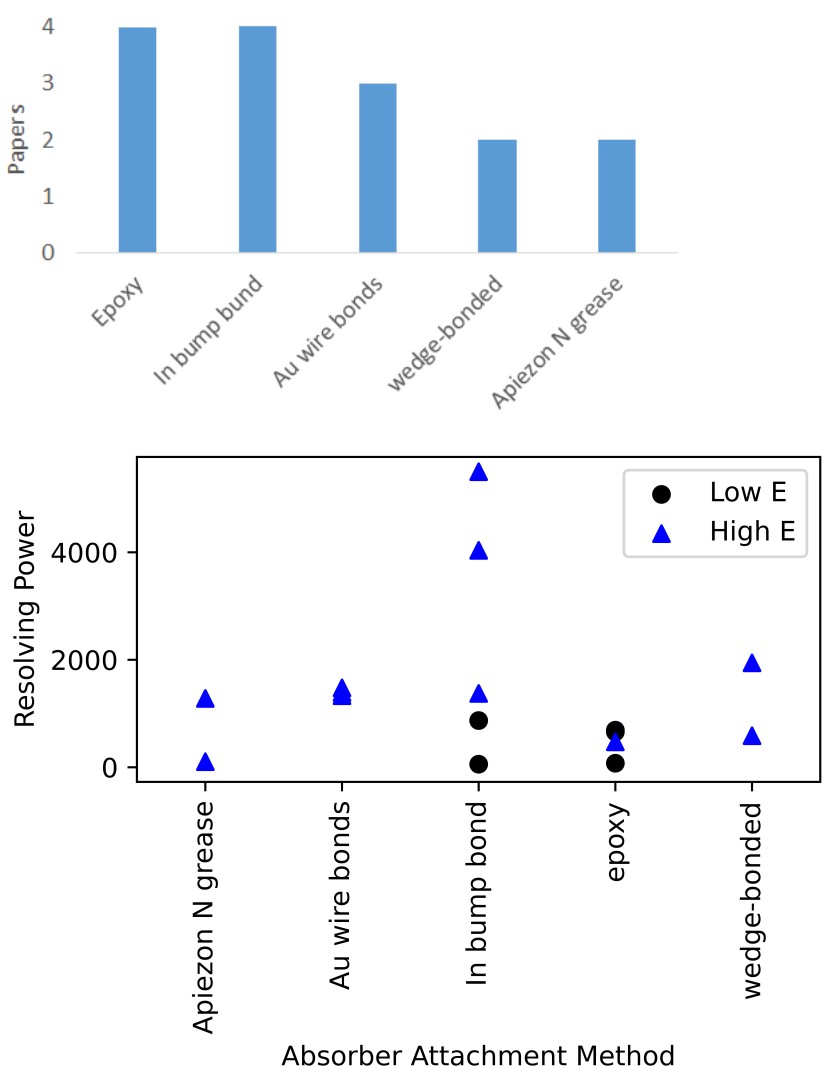

**Figure 6.** (*Top*) Absorber attachment methods used in Decay Energy Spectroscopy. (*Bottom*) The resolving power can be seen as a function of absorber attachment method for both high energy DES and low energy DES measurements.

Of the various deposition methods (see Figure 7), ion implantation has been shown to achieve excellent results (one of the best low energy measurements [17]), but requires dedicated facilities and high efficiency throughput can be a challenge, making it a less desirable option for source material that might be sparse or inhomogenous or unknown. Electroplating has shown to produce excellent results (the best low energy measurement [35]), but this is not possible for all materials and the electrodeposition process varies by element. Autodeposition or self-deposition is a subset where no electric current need be applied because the material will spontaneously deposit on the substrate. This is possible with Po and also produces superior results (the second best results with high energy measurements [55]). The preferred method for ease of mass-production and ability to deposit any radionuclide is the dried drop method. In this method, the radionuclide is usually a dissolved salt within the solution or an organic molecule with carrier. The liquid is deposited onto the substrate—usually a Au foil—and allowed to dry. Upon drying, it has been found that the salts within the solution have dried down along with the radionuclide and the mass of that salt is significantly more than the mass of just the radionuclide. The resultant residue degrades energy resolution as the energy from the decay is thermalized both within the residue and in the absorber matrix. The resolving power as a function of deposition method is given in Figure 7.

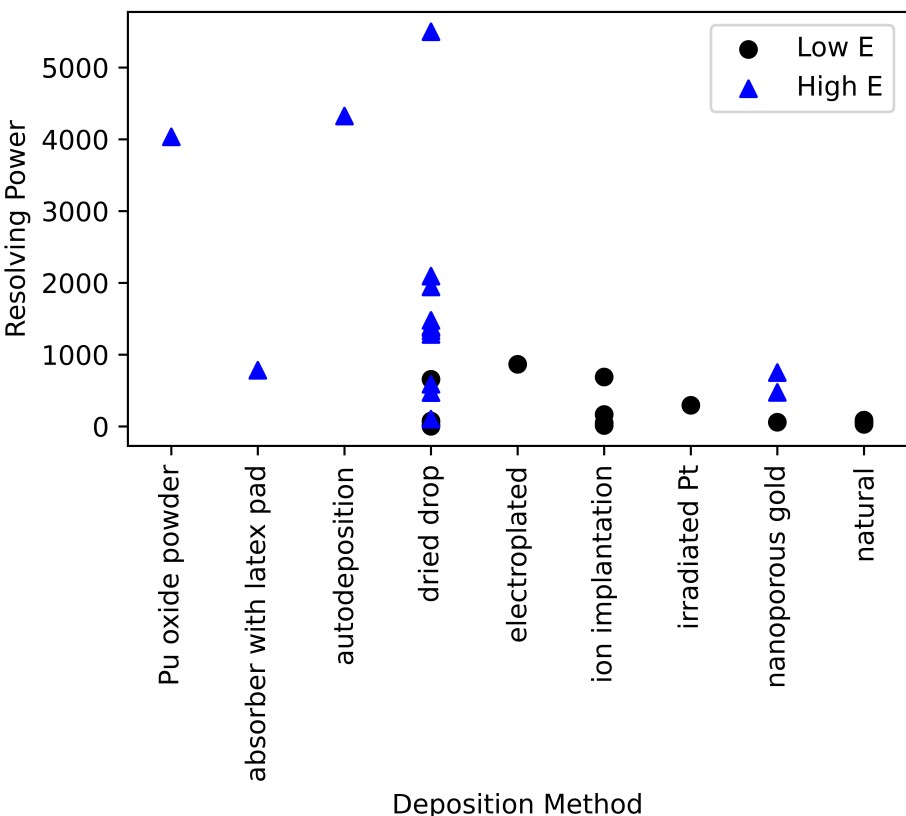

**Figure 7.** Detector resolving power as a function of deposition method.

The effects of improper deposition techniques are degraded energy resolution, shifts in peak energy [9,16], long tailing [9], spectral distortion [68], and even peak splitting [9,35]. Several different approaches are undertaken to address the problem of the crystalline deposits in the dried droplet method. One approach is to reduce the size of the crystalline deposits by using automatic microdrop dispensers [35,67]. Reducing the thickness of the deposit improves energy resolution [69]. A second approach is to deposit the material into a nanoporous Au, so the pores control the size of the crystalline deposits. The challenge here is creating nanoporous Au with small enough pores. A third approach to control the size of the crystalline deposits is using a removable patterned latex pad to control where the material is dried and then removing the latex pad [64]. These methods have produced modest improvements. Other approaches to improve the results after the initial deposition are diffusion welding and mechanical alloying. The samples that were diffusion welded after deposition are highlighted in Figure 8 and the samples that were subsequently mechanically alloyed are highlighted in Figure 9. Mechanically alloying has produced the best high energy DES result to date [58], but had a modest effect on improving a low energy DES measurement of a dried deposit in nanoporous gold [35].

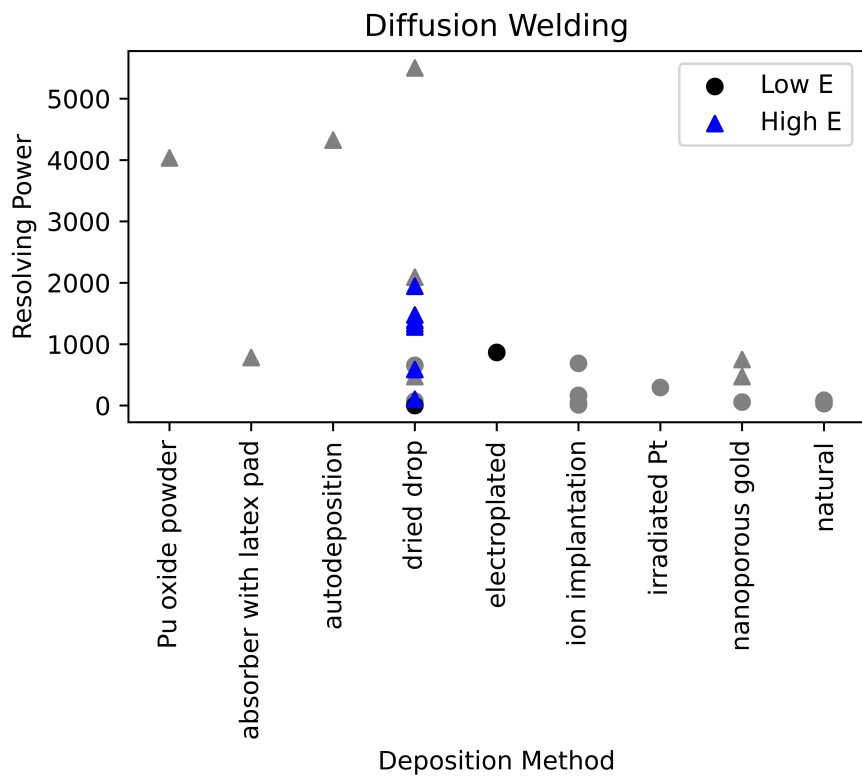

**Figure 8.** Detector resolving power as a function of deposition method. The effects of diffusion welding are highlighted here.

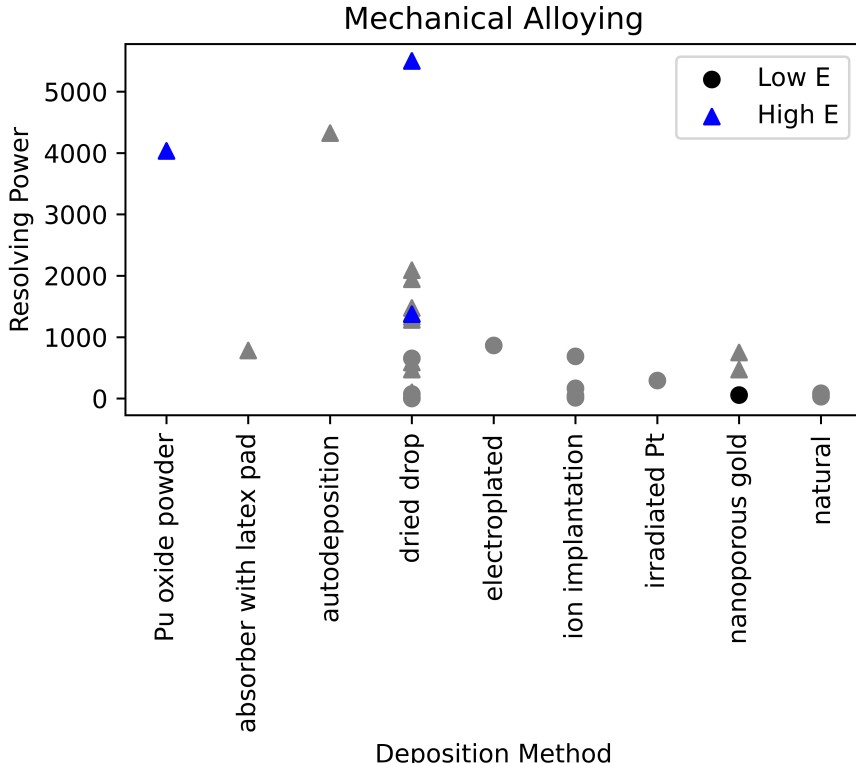

**Figure 9.** Detector resolving power as a function of deposition method. The effects of mechanical alloying are highlighted here.

## 5. Discussion

The burgeoning field of Decay Energy Spectroscopy is useful in a variety of application areas including neutrino physics, metrology, safeguards, medical radionuclides, and environmental samples. Achieving consistent, high energy resolution measurements has proven to be a challenge addressed in a number of different ways, depending on the application. Excellent results have been achieved with multiple detector types. Sample preparation has proved to be key to improving results, with the best measurements using mechanically alloying on a dried deposit, electroplating or autodeposition, or ion implantation. Since not all radionuclides can be electroplated and ion implantation is expensive and inefficient, the best method in the literature for high energy resolution is mechanically alloying or kneading the sample. It appears that diffusion welding a dried droplet sample does not improve the achieved resolution. Excellent results have been achieved with both In bump bond attachment methods as well as epoxy.

As high energy resolution DES measurements become more consistent, the dominating energy resolution term from energy loss within the sample deposits will become consistent and measureable. Similar to work done for $\alpha$ spectroscopy [7], further work should be done to quantify the effects of various factors on the achieved energy resolution of DES detectors to determine why low energy DES measurements have yet to exceed the 1000 threshold on resolving power, while high energy DES measurements are able to achieve a resolving power of 5000.

**Funding:** This compilation of these results was partially funded by the DOE/NNSA's Office of International Nuclear Safeguards.

**Acknowledgments:** The author would like to thank the Los Alamos National Laboratory low temperature detector team for their invaluable support, in particular Mark Croce and Mike Rabin. Sincere thanks also to colleagues Philipp Ranitzsch, Martin Loidl, and Matias Rodrigues for technical discussions and helping me understand the broader context of DES work.

**Conflicts of Interest:** The author declares no conflict of interest. The funder had no role in the design of the study; in the collection, analyses, or interpretation of data; in the writing of the manuscript, or in the decision to publish the results.

## Abbreviations

The following abbreviations are used in this manuscript:

| | |
|---|---|
| BEFS | Beta Environmental Fine Structure |
| CEA | French Atomic Energy Commission |
| CUORE | Cryogenic Underground Observatory for Rare Events |
| DES | Decay Energy Spectroscopy |
| ECHo | Electron Capture $^{163}$Ho experiment |
| HOLMES | Holmium neutrino experiment |
| KRISS | Korea Research Institute of Standards and Science |
| LANL | Los Alamos National Laboratory |
| MARE | Microcalorimeter Arrays for a Neutrino Mass Experiment |
| MMC | Metallic Magnetic Calorimeter |
| NTD | Neutron Transmutation Doped |
| NuMECS | Neutrino Mass via $^{163}$Ho Electron Capture Spectroscopy |
| TES | Transition Edge Sensor |

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
