# Peer review of "Low Temperature Microcalorimeters for Decay Energy Spectroscopy"

_applsci, doi:10.3390/app11094044_

Round 1

Reviewer 1 Report

The paper presents a review of applications and methods of low temperature micro-calorimeters. The resolving power achieved with different classes of thermal sensors and with different sensor-to-absorber couplings are discussed.
This is a short and interesting overview of the advances of the experimental techniques developed for energy spectroscopy with micro-calorimeters.
I recommend this short review for publications after minor changes.

One general comment: while reading the manuscript it is not clear weather the author is reviewing DES methods for low-temperature calorimeter/bolometer in general (macro and micro), or just the ones for micro-calorimeter applications. In fact, in the manuscript it is rarely mentioned that the main focus is on micro-calorimeters. As a mater of fact, the term "micro-calorimeter" is not even used as search term for their online studies. At this point a question may arise: are the 40 publications (result of the Google-scholar search) including DES with macro-calorimeters (such as: C. Cozzini et al., Phys.Rev.C 70 (2004) 064606; P. de Marcillac et al., Nature volume 422, pages 876–878 (2003); N. Casali et al., J. Phys. G: Nucl. Part. Phys. 41 075101 (2014), ...)?
I would suggest to include in the introduction some sentence specifying that the review is mainly focusing on micro-calorimeter applications.

Few minor comments:

line 56: why did you omit thermistors (like Ge-NTDs)?

Fig 1. It could be beneficial to explicit the definition of Low/High E also in the caption.

line 127: I suggest to give some motivation for not including the bolometer research (e.g. mostly employed for macro-calorimeters?).

Fig. 6 a label on the y-axis is missing.

Author Response

>One general comment: while reading the manuscript it is not clear weather the author is reviewing DES methods for low-temperature calorimeter/bolometer in general (macro and micro), or just the ones for micro-calorimeter applications. In fact, in the manuscript it is rarely mentioned that the main focus is on micro-calorimeters. 
This is a good point and this has been changed in the first paragraph.

>As a mater of fact, the term "micro-calorimeter" is not even used as search term for their online studies. 
I actually did use the search term microcalorimeter, but it also turned up too many papers that were not relevant, so I chose not to use this even though microcalorimeter measurements was a guiding principle for this paper.

>At this point a question may arise: are the 40 publications (result of the Google-scholar search) including DES with macro-calorimeters (such as: C. Cozzini et al., Phys.Rev.C 70 (2004) 064606; P. de Marcillac et al., Nature volume 422, pages 876–878 (2003); N. Casali et al., J. Phys. G: Nucl. Part. Phys. 41 075101 (2014), ...)? I would suggest to include in the introduction some sentence specifying that the review is mainly focusing on micro-calorimeter applications.
This is a good point and microcalorimeter has been added to the abstract and introduction, since macrocalorimeters were implictly excluded but not explicitly stated.

>line 56: why did you omit thermistors (like Ge-NTDs)?
Because this came up with too many irrelevant papers and the ones that I did find were bolometers. This search was done, but did not add any new papers to the ones I had already found. I have added a statement indicating that thermistor and microcalorimeter were considered as search terms.

>Fig 1. It could be beneficial to explicit the definition of Low/High E also in the caption.
Added.

>line 127: I suggest to give some motivation for not including the bolometer research (e.g. mostly employed for macro-calorimeters?).
This is indeed the case and a statement making this explicit is now included.

>Fig. 6 a label on the y-axis is missing.
Fixed.

Reviewer 2 Report

The manuscript “Low Temperature Microcalorimeters for Decay Energy Spectroscopy” presents a review of the literature available on the subject of Low temperature micro-calorimeters applied to the measurement of embedded radioisotopes. The review is focused on the energy resolution achievable with this technique as a function of various choices of detector, absorber, and sample preparation method. As the results on the energy resolution for these detectors versus the predicted ones are strongly dependent on the above choices, the review discusses probable causes and solutions found in the body of literature. 

The study is conducted on a total of 40 papers, resulting from a search of the existing literature and a subsequent scrutiny. A database is then compiled of metadata and data from the analysed paper. Moreover, a careful recount of the history of Decay Energy Spectroscopy is provided, which I found interesting.

The results are presented in a series of plots that show the resolving power as a function of the various choices of detector, absorber material, attachment and deposition method. The energy resolution does not appear to be a function of detector type. Excellent results have been achieved with multiple detector types and sample preparation seems to be key to improving results. 

In the discussion, it is highlighted which of the methods have provided the best results for low and high energy measurements respectively.

The manuscript is well written and accessible to non experts, it makes ample reference of the appropriate literature on the field. In my opinion it is a fairly complete review of the energy resolution achieved with micro-calorimeters. Throughout the paper, there is little attempt to discuss and understand what are the reasons behind a low or high resolving power for a given method, however this is probably out of the scope of a review and the paper is of sure interest to researchers in the field, as it compiles the  various sample preparation methods in the light of energy resolution performance.  I consider the manuscript suitable for publication in Applied Sciences in the present form.

Author Response

>Throughout the paper, there is little attempt to discuss and understand what are the reasons behind a low or high resolving power for a given method, however this is probably out of the scope of a review and the paper is of sure interest to researchers in the field, as it compiles the  various sample preparation methods in the light of energy resolution performance.

As noted, this is out of the scope of this review. This review does not aim to provide reasons beyond what is included in the literature.

Reviewer 3 Report

In the article "Low-Temperature Microcalorimeters for Decay Energy Spectroscopy", the author gives a clear and comprehensive overview of the different experimental efforts to measure the energy spectrum of several decays. The article is well written and pleasant to read, the only general comment is that it could be more quantitative when mentioning resolutions and other performances of the different detectors/experiments. I recommend publishing the paper after the following minor revisions. 

Specific comments:

Line 16: Missing space after the reference n.11.

Figure 1: You should specify what do you mean by "low-energy" and "high-energy" in the caption. 

Line 126: You can quote a paper from the CUORE collaboration, such as  "High sensitivity neutrinoless double-beta decay search with one tonne-year of CUORE data" (https://arxiv.org/pdf/2104.06906.pdf), which seems to be the most recent one.

Lines 132-141: Please organize better this paragraph. In particular, you should introduce "the initial Fe-55 measurement".

Line 148: Report the achieved numbers when you write "poor resolution".

Line 302: Neutron Transmutation Doped

Ref. 58: Please check the title

Author Response

> Line 16: Missing space after the reference n.11.

Fixed.

>Figure 1: You should specify what do you mean by "low-energy" and "high-energy" in the caption. 

Fixed.

>Line 126: You can quote a paper from the CUORE collaboration, such as  "High sensitivity neutrinoless double-beta decay search with one tonne-year of CUORE data" (https://arxiv.org/pdf/2104.06906.pdf), which seems to be the most recent one.

Added.

>Lines 132-141: Please organize better this paragraph. In particular, you should introduce "the initial Fe-55 measurement".

This paragraph and the following paragraph were reworded to be clearer as well as add other important information on the difference in energy thermalization between x-rays and electrons.

>Line 148: Report the achieved numbers when you write "poor resolution".

Numbers were not reported in this reference, but I have included the best resolution that is reported.

>Line 302: Neutron Transmutation Doped

Fixed.

>Ref. 58: Please check the title

Fixed.